# A Time-Varying Model for Predicting Formaldehyde Emission Rates in Homes

**DOI:** 10.3390/ijerph19116603

**Published:** 2022-05-28

**Authors:** Haoran Zhao, Iain S. Walker, Michael D. Sohn, Brennan Less

**Affiliations:** Residential Building Systems Group and Indoor Environment Group, Lawrence Berkeley National Laboratory, Berkeley, CA 94720, USA; iswalker@lbl.gov (I.S.W.); mdsohn@lbl.gov (M.D.S.); bdless@lbl.gov (B.L.)

**Keywords:** formaldehyde, indoor air quality, emission rate, new homes, field measured data, temperature, humidity, modeling, simulation

## Abstract

Recent studies have succeeded in relating emissions of various volatile organic compounds to material mass diffusion transfer using detailed empirical characteristics of each of the individual emitting materials. While significant, the resulting models are often scenario specific and/or require a host of individual component parameters to estimate emission rates. This study developed an approach to estimate aggregated emissions rates based on a wide number of field measurements. We used a multi-parameter regression model based on previous mass transfer models to predict formaldehyde emission rate for a whole dwelling using field-measured, time-resolved formaldehyde concentrations, air exchange rates, and indoor environmental parameters in 63 California single-family houses built between 2011 and 2017. The resulting model provides time-varying formaldehyde emission rates, normalized by floor area, for each study home, assuming a well-mixed mass balance transport model of the home, and a well-mixed layer transport model of indoor surfaces. The surface layer model asserts an equilibrium concentration within the surface layer of the emitted materials that is a function of temperature and *RH*; the dwelling ventilation rate serves as a surrogate for indoor concentration. We also developed a more generic emission model that is suitable for broad prediction of emission for a population of buildings. This model is also based on measurements aggregated from 27 homes from the same study. We showed that errors in predicting household formaldehyde concentrations using this approach were substantially less than those using a traditional constant emission rate model, despite requiring less unique building information.

## 1. Introduction

Formaldehyde emission rates have most commonly been estimated as fixed values based on materials and house characteristics that are invariant with temperature, humidity or air change rate [1,2]. However, previous field measurements and modeling studies have shown that these environmental factors influence indoor formaldehyde emissions. The objective of this study was to develop an improved calculation procedure to estimate the emission rate of formaldehyde for modeling indoor air quality in residential buildings. Rather than a fixed emission rate, we developed an emission rate model that varies in time depending on environmental parameters. The model-development procedure was based on using measured field data to estimate emission rates and to correlate the emission rate with various commonly known indoor parameters: temperature, humidity, ventilation rate, and floor area. The intent was that the emission rate estimates could be used to determine formaldehyde concentrations in indoor air in homes under dynamic environmental and ventilation conditions. Some previous studies have used similar physical models, but were based on coefficients from test chamber samples rather than emissions in real homes requiring considerable assumptions, to convert emissions from samples to emissions from all the sources in a home. In order to remove these assumptions, the current study used measured data from a group of homes to investigate home-to-home emission rate variability. The value of accurate, time-varying emission estimates for formaldehyde in dwellings is that we can improve our ability to predict changes in formaldehyde exposures associated with changes in the building code, dwelling ventilation rates or controls.

Estimation of formaldehyde concentrations in homes is important, because indoor exposures are associated with substantial public health burdens, as quantified by Disability Adjusted Life Years (DALYs) [3]. Formaldehyde is found in many common building materials, such as engineered wood products, cabinetry, and flooring. Acute exposure to formaldehyde can cause nose, eye, and throat irritation [4,5]. Indoor chronic exposure to formaldehyde can cause respiratory symptoms and cancer [6,7].

Previous studies have found that formaldehyde concentrations in new and existing residences routinely exceed health relevant thresholds. In a study of 105 California homes built in 2002–2005 [8], formaldehyde concentrations measured over 24-h exceeded the California Office of Environmental Health Hazards Assessment (OEHHA) chronic and 8-h reference exposure levels of 9 µg/m^3^ (7.3 ppb) [9] in 98% of homes, and exceeded the World Health Organization 30-min exposure guideline of 0.1 mg/m^3^ [4] in 5% of the homes. In a more recent study in California of newly built homes with low-emitting materials, weekly average formaldehyde concentrations exceeded the California OEHHA chronic and 8-h reference exposure levels in all 68 homes [10]. In another large-scale study of 352 existing California homes, the average indoor formaldehyde concentration was lower (14 µg/m^3^), but 95% homes still exceeded the California OEHHA chronic and 8-h reference exposure levels [11]. In a study investigating 398 existing US homes, the mean indoor formaldehyde concentration was 21.6 µg/m^3^ [12]. Two recent surveys conducted in Canada found formaldehyde concentrations (averaged over 24-h) exceeded the Health Canada residential indoor air quality 8-h exposure guideline of 50 µg/m^3^ [13] in 16 out of 59 homes (27%) in Prince Edward Island [14], and in 11 out of 96 homes (11%) in Quebec City [15].

There are two key engineering control measures for limiting formaldehyde concentrations in residences: source control through use of low-emitting materials and ventilation with outside air.

Formaldehyde emissions from engineered wood products were first regulated in the US by the State of California beginning in 2007. National regulations based on the California requirements (Formaldehyde Standards for Composite Wood Products Act of 2010—Code of Federal Regulations 40 CFR Part 770) [16] were legislated in 2010 and came into force in March of 2019. These standards limit formaldehyde emissions by prescribing maximum allowable equilibrium concentrations measured in laboratory chamber tests of product samples under standard conditions. The effectiveness of these regulations in reducing formaldehyde concentrations has been demonstrated in field studies [17]. However, the emission rates are not measured or regulated directly by the standard. The actual emission rate of the emitting materials in homes is dominated by the maximum allowable equilibrium concentration, but also affected by environmental conditions, such as temperature and relative humidity, which will be described in the later sections. The whole house equilibrium concentration is highly variable because the type and quantity of the materials in homes remain unknown. Thus, the model we developed in this study will focus on a specific group of homes with low- emitting materials. The whole house equilibrium concentration was estimated as a coefficient to represent house to house variance.

Ventilation with outside air can reduce the concentration of formaldehyde in residences via dilution and removal. Quantitative studies have demonstrated how formaldehyde concentrations decrease as air change rates increase cross-sectionally for populations of homes [17,18,19,20]. Based on formaldehyde emission factors, one study estimated that an air change rate (ACH) of 0.5 h^−1^ would maintain formaldehyde concentrations below 50 ppb in typical new North American houses [21]. A study in Canadian homes found that an ACH of 0.35 h^−1^ appears sufficient to ensure a formaldehyde concentration lower than the 50 µg/m^3^ Health Canada guideline in most homes [15].

Increasing ventilation rate would increase the concentration gradient between material air surface layer and the room air, leading to an increment of whole house formaldehyde emission rate, as suggested by previous studies [17,22,23,24,25,26]. Specifically, ventilation has been shown to increase formaldehyde emissions, while still providing a net-reduction in indoor concentrations. Hult et al. reported measurements in nine new homes in which they systematically achieved three different ventilation rates [17]. As ventilation rates were increased, they compared observed reductions in formaldehyde concentrations versus expected reductions based a fixed emission rate assumption. They found that up to 60% of the benefit of increased ventilation (assuming fixed emissions) was lost due to corresponding increases in formaldehyde emission rates. Liu et al. performed a time-resolved assessment of VOC emission rates (including formaldehyde) in a Northern California residence, and they found that emission rates increased with household ventilation rates and with temperature [25]. Offermann et al. tested three ventilation rates in a CA home (0.21, 0.41 and 0.64 h^−1^), and observed formaldehyde emissions increase from 17 to 24 to 31 µg/m^3^/h [26].

Several studies have measured changes in formaldehyde concentrations associated with temperature and humidity conditions, from which changes in emission rates are inferred. Andersen et al. measured an increase in temperature of 7 °C doubled the formaldehyde equilibrium concentration in a chamber and the change of 40% relative humidity also doubled the formaldehyde concentration [27]. Salthammer et al. measured an increase of 6 μg/m^3^ per Celsius degree increase in temperature and 2 μg/m^3^ for every 1% increase in relative humidity [19]. Poppendieck et al. found almost a factor of 2.6 increase in VOCs for an 8 °C increase in temperature [22]. Another study measured four homes during winter and summer and found changes of about three to five ppb per °C [23]. A study measuring formaldehyde concentrations in environmental chamber tests [28] found formaldehyde concentrations were positively correlated with temperature and absolute humidity, but were poorly correlated with relative humidity.

Studies using material mass diffusion transfer analytical models to predict the volatile organic compound (VOC) emission rates from multiple building materials have demonstrated that VOC emission rates are dominated by three factors: the initial emittable concentration (C_0_), a mass diffusion coefficient for the compound in building materials (D_m_), and the material-air partition coefficient (K_p_) [29,30,31]. Chamber studies have shown that K_p_ and D_m_ are strongly influenced by temperature [32,33,34] and C_o_ is driven by both temperature and humidity [35]. By including the factors mentioned above, previous studies have used measured emission rates of material samples from test chambers to determine the parameters for mass diffusion transfer (C_0_, D_m_, and K_p_), and, in turn, to develop physics-based emission models to predict the emission rate for materials, such as a particleboard, plywood paneling, or hard wood piece with UF coating [29,30,31]. The studies investigating physical models have provided referenced parameters for estimating the mass transfer and the relationship between these parameters and environmental conditions. However, to utilize the chamber measured parameters to predict a whole house emission rate, the quantities of each type of emitting materials in the house need to be characterized, including structural materials, furniture, cabinetry, and other indoor sources. For example, in a previous case study in an unfurnished net-zero house [36], the investigator characterized the surface area of the emitting surfaces and determined the corresponding mass transfer dynamic parameters (C_0_, D_m_, and K_p_) from previous chamber studies. The whole house VOC emission rate was estimated using a physics-based model combined with measured temperature and humidity. A similar study by Bourdin et al. used a simplified surface mass transfer model to estimate formaldehyde emission rate in a classroom [37]. The mass transfer coefficient (D_m_) was estimated using an empirical equation and the concentration at the surface layer of each type of material was measured in a chamber. With an accurately measured surface area of each piece of emitting material in the room, the dynamic formaldehyde emission rate was estimated. Generally, it is very difficult and impractical to measure the required input data for the physics-based models for each emitting material in each home. The characterization of emitting materials in real homes during field test is impracticable due to limit amount of time, specifically the mixture of quantities and material types present in any given home.

Data-driven approaches, such as partial linear regression, machine learning, and deep learning, have been previously developed to predict indoor air pollutant concentrations such as CO_2_ and PM_2.5_ [38]. Those data-drive approaches were expanded for predicting formaldehyde emission rate and/or concentrations for certain materials in the chamber or a whole building in recent studies. Akyüz et al. presented an implantation of artificial neural networks (ANN) for modeling the formaldehyde emission from particleboard based on manufacturing variables, including wood-glue moisture content, density of board, and pressing temperature [39]. Ouaret et al. developed an approach using Fourier transform and two nonlinear model: threshold autoregressive (TAR) and Chaos dynamics models to forecast the formaldehyde concentration 12 h ahead in a regularly occupied office with diurnal pattern [40]. Zhang et al. recently applied an artificial neural networks (ANN) approach to predict gas-phase VOC concentrations from four kinds of furniture in a chamber [41]. The ANN approach used VOC concentration at the previous timestep, temperature, relative humidity, and ventilation rate as inputs for training the model, and the method was validated by predicting predict VOC concentrations for different environmental conditions. Zhang et al. also developed another approach using deep learning model and tested it in an occupied classroom [42]. Similarly, Mohammadshirazi et al. also used a LSTM deep learning approach to predict formaldehyde concentration in an occupied office based on historical measured data [43] and compared to other three forecasting models: rolling average, Random Forest, and Gradient Boosting. The data-driven methods do not require detailed mass transfer parameters of the emitting materials, but the approaches typically need massive data for training and the approaches have not been applied to any residential buildings, where the environmental and occupancy pattern are more complex than commercial buildings. In addition, it is unknown how well these models trained for an individual building or room would predict emission rates or concentrations in other spaces. In our study we want to develop a model that could be applied beyond an individual home where the measurements were made. Therefore, we developed an empirical model derived from physics-based model in order to bypass characterizing mass transfer parameters as well as involving the emission rate variability due to environmental conditions.

In indoor environments, formaldehyde mass transport from building materials has been modeled by assuming the concentration in a thin layer of air near the emitting material (*C_eq_*) is in equilibrium with the contaminant concentration in the surface layer of the storage medium (*C_material_*). *C_material_* remains constant because internal transport within the material is rapid enough to replenish the surface layer as it emits into the air above the surface. At any given moment, the formaldehyde emission is governed by transport from air surface layer to bulk air at a constant rate (*k*), as shown in Equation (1) [17,31,44,45,46].
(1)$$E=k\left(C_{eq}-C_{in}\right)A_{m}$$
where *E* is the whole-house formaldehyde emission rate (μg/s) at certain temperature and humidity; *C_eq_* is the concentration within the air surface layer, which is equal to the bulk air concentration when the ventilation rate is zero (μg/m^3^) at that condition; *C_in_* is the concentration in the bulk air (μg/m^3^) (i.e., the indoor formaldehyde concentration); *k* is the mass transfer constant (m/s); and *A_m_* is the effective surface area of the emitting materials indoors (m^2^). The values of *k, C_eq_*, and *C_in_* are treated as whole-house values that include all the emitting surfaces in the home. This simplified concentration-dependent emission model has been validated in chamber studies [44,45], and the implications of indoor formaldehyde concentration on emission rate were discussed in a previous study [46]. Under this model, the changes in temperature and humidity will influence *C_eq_*, which has been investigated in previous studies [27,47,48]. The indoor air exchange rate will influence *C_in_*. Changes of indoor temperature and air velocity caused by changing air change rate will also influence the mass transfer rate (k), but this is assumed to be negligible in this study, because the variations of indoor temperature and surface air velocity are small.

The focus of this paper is to develop a simplified emission model for predicting formaldehyde emission rates in homes that vary with the key variables explored above. This is a top-down approach that does not rely on the availability of highly specific details about emitting materials in each home. Instead, we focused on estimating whole-house emission rates across a group of sample homes using three predictor variables: temperature, humidity, and air change rate. We used field measured, time-resolved formaldehyde concentrations together with coincident temperature and relative humidity in 63 California single-family homes built in 2011–2017 with low-emitting materials (i.e., materials regulated by the California Air Resources Board). Estimates of time-resolved ventilation rates were used in a mass balance to estimate the time-varying formaldehyde emission rates. A multivariate regression was used to develop a predictive model for emission rate as a function of temperature, humidity, and air change rate. The resulting emission model was evaluated by using it to predict the concentrations in the same homes used to generate the emission model, and then comparing the resulting predicted concentrations against those measured in the field. This evaluation was performed for individual model coefficients specific to each home, as well as for coefficients averaged over the cohort of homes.

## 2. Materials and Methods

### 2.1. Data Collection Overview

The data used to estimate formaldehyde emission rates were collected from a recent field study of ventilation and indoor air quality in new California homes. The Healthy, Efficient New Gas Homes (HENGH) study [49] collected data in 2016–2018 in 70 single-family, detached houses that were constructed between 2011 and 2017. These homes were built using composite wood products required in California to have low formaldehyde emissions. All homes had dwelling unit mechanical ventilation systems installed to meet state building code requirements. The homes also had natural gas cooking appliances with venting range hoods and bathroom exhaust fans. Each home was monitored for a six-to-nine-day period. Residents were asked to keep windows closed and the dwelling unit mechanical ventilation system operating. This allowed us to make good estimates of the ventilation rate, because the homes were dominated by the air flow through mechanical ventilation systems, for which airflows and operation were measured and recorded in the study. We included estimates of natural infiltration, based on air leakage tests of the homes, local weather conditions (from the nearest publicly available weather station) and the enhanced ventilation model from ASHRAE Handbook of Fundamentals [50] and Walker and Wilson [51]. The total ventilation rate combining the mechanical fan and natural infiltration flows was determined using the superposition method from ASHRAE Standard 62.2 and a previous study [52].

Measurements in each home included time-integrated indoor and outdoor formaldehyde concentrations, and additional time-resolved measurements were made indoors. The outdoor formaldehyde concentrations were measured using SKC Umex-100 passive samplers at each site to obtain an average concentration for the whole monitoring period. The time-resolved indoor formaldehyde concentrations were measured at 30-min or 60-min intervals using Shinyei/Graywolf FM-801 photoelectric photometry meters deployed in the living rooms and master bedrooms in most of the homes. The averaged indoor concentrations calculated using the real-time meters were compared against UMEx-100 passive samplers deployed at the same location with same duration in 66 test homes that had both types of measurements. Results showed considerable scattering between co-located real-time formaldehyde meters and passive samplers, as shown in Figure 1. The weekly averaged indoor concentrations measured by real-time meters compared to the passive samplers, had a negative bias of 2% and root mean square difference (RMSE) of 6.9 μg/m^3^. The temperature and relative humidity were also measured in the living rooms and master bedrooms in these houses using ExTech CO_2_ monitors at 1-min intervals.

The airflows of bath and laundry exhaust fans in each home were measured using a TEC Exhaust Fan Flow Meter. Range hood airflows were measured using a balanced-pressure flow hood method using a TEC Minneapolis Duct Blaster [53]. The operation of these fans was monitored at 1-min intervals using a logging anemometer (Digisense WD-20250-22) placed at the air inlet or using a motor on-off logger (Onset HOBO UX90-004) placed close to the motor. The air leakage of the building envelope and the forced air heating/cooling system were measured with the DeltaQ test (ASTM-E1554-2013, Method A) using a TEC Minneapolis Blower Door System with DG-700 digital manometer. The test also quantifies air leakage of the forced air heating/cooling system to outside of the living space under normal operating conditions. Building envelope air leakage was converted to air changes per hour at 50 Pa indoor-outdoor pressure difference (ACH_50_) using the estimated home volume. Not all of the 70 homes in the original study were included in the analysis. Two houses had instrumentation failure, four homes did not have measured envelope air leakage (so we could not estimate air change rates), and one did not have temperature or relative humidity (*RH*) measured, leaving 63 homes to be evaluated.

### 2.2. Time-Resolved Formaldehyde Emission Rate Calculation

For a well-mixed home, a mass balance can be used to describe the indoor formaldehyde concentrations, as shown in Equation (2). A discretized version of Equation (2) is shown in Equation (3).
(2)$$\frac{dCin}{dt}=aCout-aCin+\frac{E}{V}$$
(3)$$E_{t}=\frac{V}{dt}\left(Cin_{t+1}-Cin_{t}\right)+a_{t}Cin_{t}V-a_{t}VCout$$
where *E_t_* is the formaldehyde net emission rate at the time step (μg/s); *dt* is the time step (1 h = 3600 s); *Cin_t_* and *Cin_t+_*_1_ are formaldehyde concentrations at this time step and one-time step after (μg/m^3^); *Cout* is the average outdoor formaldehyde concentration (μg/m^3^); *V* is the total volume of the house (m^3^); and *a_t_* is the air exchange rate (AER) (1/s). Indoor formaldehyde is both absorbed as well as emitted (in a reversible way) by building surfaces [54], therefore, in our method, the time-resolved emission rates in each home were the net-emission rate, representing both desorption and absorption processes. For most of the time, the net-emission was positive, because the combined emission and desorption were much larger than adsorption.

A key assumption of this approach is that we can use a single concentration for the whole home, and that the concentration is conditional to the whole home ventilation rates in our calculations. To investigate the validity of this approach, the well-mixed condition for formaldehyde in the test houses was evaluated by comparing the weekly average formaldehyde concentrations measured in the master bedroom and in the living room of each home, as shown in Figure 2. A linear regression fit was performed to the weekly averaged concentrations, showing reasonable correlation (R-squared of 0.64, a slope of 1.05 and an intercept of 2.4 μg/m^3^). The RMSE was 7.1 μg/m^3^. Given the instrument accuracy is 4.9 μg/m^3^ (4 ppb) or 10% of the reading (whichever is larger), these results indicate that the assumption of uniform formaldehyde concentrations was reasonable overall, but the situation varies home-by-home. We also calculated the absolute difference of the hourly measured formaldehyde concentrations between the master bedroom and living room at each home. The average absolute difference across all homes was 5.3 μg/m^3^ (4.3 ppb), which is close to the instrument accuracy. The mean absolute differences between bedroom and living room were larger than the instrument accuracy in 21 homes. Of these 21 homes, 19 had two-stories. These results indicate that our assumption of uniform concentration within a home is generally acceptable.

These concentrations are lower than those measured in new California homes built in 2002–2005 in a similar study [8], where the mean indoor concentration was 43 μg/m^3^. They are also lower than the formaldehyde concentrations measured in studies from more than ten years ago, such as 32.2 μg/m^3^ from 162 French homes [55] and about 30 μg/m^3^ from 96 Canadian homes [15]. Possible reasons for the lower concentrations in this set of homes is that the sample homes were all built with lower-emitting materials and mechanical ventilation as required by California building regulations.

### 2.3. Predictors for Formaldehyde Emission Rate

Formaldehyde is emitted from building materials, fittings, and furnishings, therefore, the emission rate scales with the quantities of these elements in a home. To be useful for future modeling purposes, the emission rates need to be normalized to account for this. A direct approach is to assume that the quantity of emitting materials scales with floor area/volume and to use floor area to normalize the emission rates. It might be that this is not an exact correlation, because the sources of formaldehyde may not scale exactly with floor area. For example, a larger house might not have proportionally more furnishings, cabinetry, etc. Furthermore, formaldehyde emissions are associated with effective surface area that is emitting formaldehyde, and the formaldehyde content of the materials present in the home varies greatly between homes. For the purposes of this study, we performed a simple analysis to verify that normalizing by floor area is a reasonable assumption, which is discussed in the results section.

Previous studies have demonstrated that *C_eq_* varies with temperature and relative humidity [48]. The temperature and *RH* varied within each home during the sampling period. These parameters also varied cross-sectionally, since the homes were sampled in different seasons, had different thermostat setpoints, different occupancies, and other changes that would impact temperature and *RH*. In order to account for the impacts of temperature and *RH* on the emission rate, an empirical mathematical equation from Myers [48] was used to normalize the equilibrium formaldehyde concentration to a value (*C_st_*) at reference condition (i.e., 25 °C, 50% *RH*), as shown in Equation (4).
(4)$$C_{eq}=C_{st}\exp\left[A\left(\frac{1}{T}-\frac{1}{298}\right)\right]\times \left[1+B\left(RH-50\right)\right]$$

For indoor environments when the variation of temperature is small, a simple linear form is often used:(5)$$C_{eq}=C_{st}\left[1+A\left(T-25\right)\right]\times \left[1+B\left(RH-50\right)\right]$$

Applying Equation (5) to Equation (1), the time-varying concentration-dependent emission model corelated with temperature and relative humidity is shown in Equation (6).
(6)$$\frac{E_{t}}{A_{f}}=kL\{C_{st}\left[1+A\left(T_{t}-25\right)\right]\times \left[1+B\left(RH_{t}-50\right)\right]-Cin_{t}\}\times H$$
where *A*, *B*, and *C_st_* are the fitted parameters. *L* is the effective emitting material loading rate in the house (*A_m_/V*, m^2^/m^3^). *T_t_* and *RH_t_* are the average temperature and relative humidity of the rooms with a formaldehyde monitor (in this case, the living room and a bedroom of each home) at the time step. Et and Cint are the emission rate and indoor formaldehyde concentration at the time step. *A_f_* is the floor area and *H* is the ceiling height. We did try using the Equation (6) and the field measured hourly formaldehyde concentrations to develop our model but the performance of this model to estimate time-resolved concentrations was dissatisfying. A main problem with this approach was that it used the concentration from a previous time step to estimate the emission rate for the current time step because the current concentration is unknown. To improve the modeling accuracy for predicting time-resolved concentrations, we developed an approach to use hourly air exchange rate as a surrogate for the concentration in the bulk room air, which has been used in previous studies [14,38,39,40].

In our measured data, the indoor formaldehyde concentrations were sampled hourly, and the differences between consecutive hours (i.e., *Cin_t+_*_1_ − *Cin_t_*) were typically very small. Similarly, the temperature and *RH* varied slowly with time. Air exchange rate could change suddenly due to mechanical ventilation operation, but the duration of the fan usage was typically short, leading the hourly variation of the total air exchange rate to be relatively small. Therefore, we assume that the measured formaldehyde concentration at each hour can be considered to be a pseudo-steady-state concentration under the corresponding temperature, *RH*, and AER conditions. The calculated hourly emission rate was, therefore, also a pseudo-steady-state emission rate. For a given well-mixed home, the steady-state indoor concentration (*Cin_ss_*) under certain temperature, relative humidity, and air exchange conditions can be expressed using Equation (7).
(7)$$Cin_{ss}=Cout+\frac{E}{aV}$$

Replacing Equation (1) with a pseudo-steady-state emission rate and concentration yields Equation (8).
(8)$$E=(C_{eq}-Cout)\frac{akL}{a+kL}V$$

In our measured data, the average outdoor formaldehyde concentration is 2.2 μg/m^3^. This suggests the outdoor term is an order of magnitude smaller than the indoor term. Ignoring the outdoor term and combining with Equation (5), the hourly pseudo-steady-state emission rate was correlated to the hourly average measured temperature, *RH* with modeling coefficients (*A* and *B*) and equilibrium concentration at reference condition (*C_st_*) through a multi-parameter model, along with hourly air exchange rates, mass transport coefficient, and loading rate as other independent inputs (Equation (9)).
(9)$$\frac{E_{t}}{A_{f}}=\frac{C_{st}\times \left(1+A\left(T_{t}-25\right)\right)\left(1+B\left(RH_{t}-50\right)\right)}{\frac{1}{a_{t}}+\frac{1}{kL}}\times H$$

In the analysis of the measured data, we used one-hour averages for all the measured parameters and corresponding one-hour averaged emission rates calculated using Equation (3). This avoided the issue generated by small step changes in some parameters—such as air change rate—from one time-step to another. The one-hour pseudo-steady-state emission rate may not be appropriate in some homes settings due to sudden changes in AER, temperature, and humidity caused by occupancy activities or mechanical equipment operation. Such cases resulted in “outlier” emission rates at the time-step with sudden changes. To eliminate these sudden large changes in emission rate, we examined an option that excluded the hourly emission rates that were greater than the 95th percentile and smaller than the 5th percentile for each home prior to fitting the regression model in Equation (9). We also investigated an approach that applied an eight-hour running average to the measured formaldehyde concentration, air exchange rate, temperature and *RH*. Given the average air exchange rate across all homes was 0.35 h^−1^ and the ventilation was the only effective loss term for indoor formaldehyde, a home would generally achieve steady-state within about eight hours after any changes in emission and ventilation. The multi-parameter regressions were compared using the running 8-h and 1-h inputs. The approach was used to check whether hourly emission rates could be assumed to be pseudo-steady-state.

The constant *kL* is the product of the transport coefficient *k* and the loading factor *L*. Measurements of loading factor *L* from another study [56] were found to be relatively stable, ranging from 0.5 to 1 m^2^/m^3^, and the *k* value ranged from 0.011 to 3.6 m/h. Based on measurements in single-family and mobile homes, Myers reported *k* between 0.19 and 2.7 m/h [45]. Homes likely contain a range of formaldehyde-containing materials, but the fastest timescales (higher *kL*) will tend to dominate the effective value for a home [46]. In our model, we use the average value from Sherman and Hult which reported *kL* values from 0.05 to 0.62 in nine low-emitting US houses (most in California) with average of 0.29/h^−1^ [46]. We did attempt to also include *kL* as a fitted parameter in our model, but this resulted in unstable model values, with large variations in *kL* that also drove large changes in the other three coefficients that were out of the range of common building materials reported by Myers. This is likely caused by limited data in each home, where we only have a small range of temperature, *RH*, and AER, resulting in least square regression results with large uncertainties. Due to this result, we chose to fix the value of *kL* based on those found in the literature described above.

A multi-parameter, least square non-linear regression fit was applied to Equation (9) for each house. The independent inputs of the regression fit were the measured temperature, *RH* and air exchange rate, either by hourly step or 8-h running average. The *kL* was assumed constant. The dependent input of the regression was the floor area normalized emission rate for each home, either by hourly step or 8-h running average. The regression fits resulted in 63 sets of least squares fitted coefficients: *A*, *B*, and *C_st_* for each home. We have considered multiple statistical approaches used in the previous studies for indoor air quality model evaluation [57,58]. The commonly-used parameters for model performance evaluation include standard deviation of observations and predictions, least square slope and intercept regression statistics, Quantile–Quantile (Q-Q) plots, etc. By considering all of the statistical approaches, two criteria were selected to carefully evaluate the predictor sets:The degree of correlation and the remaining errors from the empirical multi-parameter model for predicting the *emission rate*, which were identified by calculating the r-squared and root mean square error (RMSE) for each regression;The accuracy and consistency of the predicted emission rate for estimating *indoor formaldehyde concentration*, which was evaluated by comparing the measured time-resolved formaldehyde concentrations (*C_in_*) in each home versus the estimated concentration (*Cin_est_*) calculated using the regression model’s time-varying emission rate predicted by temperature, relative humidity, and air change rates at the corresponding time step, as shown in Equation (10).
(10)$$Cin_{est,t+1}=Cin_{est,t}+\left(\;\frac{E_{est,t}}{V}-a_{t}Cin_{est,t}+a_{t}Cout\right)dt$$
where *Cin_est,t_* is the estimated formaldehyde concentration at current time; *Cin_est,t+1_* is the predicted concentration for next time step; *E_est,t_* is the emission rate predicted using the three coefficients for each home; and *dt* is the length of the time step (one-hour in this case). The accuracy of the prediction was evaluated by calculating the Normalized Root Mean Square errors (NRMSE) for all the measurements (where *N* is the number of measurements, typically 160 over the week of testing for each home), as shown in Equation (11).
(11)$$Normalized\;RMS\;Error\;\left(\%\right)=\frac{\sqrt[{2}]{\sum_{0}^{N}\frac{{\left(Cin_{t}-Cin_{est,t}\right)}^{2}}{N}}}{\bar{Cin}}$$

## 3. Results and Discussion

We begin by describing the emission rate estimates produced from our calculation procedure, and we compare these against previously reported measurements. Next, we present a summary of the regression models used to estimate emission rates and the accuracy of the predicted concentrations using those same emission estimates in each home. Finally, a generalized regression model is presented that combines the model parameter coefficients from each individual study home into a generic model for future modeling efforts.

### 3.1. Emission Rate Estimation

The mean emission rate across the 63 houses calculated using Equation (3) was 1.3 µg/s with a standard deviation of 0.6 µg/s. The distribution of weekly averaged emission rates ranked by emission rate for each home is shown Figure 3. The distribution is fairly uniform between 1 and 3 µg/s, with no significant grouping at any particular emission rate. This indicates a wide range of formaldehyde sources in these homes, even though they represent a very specific subset of homes because they were selected to be new, single-family homes from California that should be compliant with State standard requiring low formaldehyde emission products to be used in their construction.

It may be useful to normalize emission rates by floor area to account for differences in emission rates for different sized homes. There is a weak general trend that emission rate increased with floor area in the study homes, with a Spearman correlation rank of 0.34 (*p*-value < 0.01) and Pearson rank of 0.31 (*p*-value = 0.016). This suggests that normalizing the emission rate by floor area can slightly improve emission rate estimates.

The distribution of floor area normalized emission rates is shown in Figure 4 with a mean (±s.d.) of 19.6 ± 10.4 µg/h/m^2^. The value is comparable to the emission rate calculated using the concentration measured by co-located passive samplers, that had a mean of 17.4 µg/h/m^2^. This is greater than the emission rate of 6.7 µg/h/m^2^ previously reported in a home designed and constructed to be low-emitting [22], but it is lower than the mean emission rate of 23 µg/h/m^2^ measured in 13 homes in another study intended to have low-emitting materials [17]. It is also lower than the average emission rate of 29 µg/h/m^2^ reported by a previous study in 99 California homes built prior to formaldehyde emission limits for building materials and that generally did not have mechanical ventilation [8]. The value is also much lower than those in older studies, such as Hodgson et al., who reported emission rates of 45 µg/h/m^2^ for manufactured homes [59].

### 3.2. Formaldehyde Emission Rate Model and Concentration Prediction

We carefully reviewed the 63 sets of regression coefficients, and we noticed that some homes had regression coefficients of *A* < 0 or *B* < 0. We consider these to be non-physical results, because emission rates should positively correlate with temperature and *RH* according to previous studies. Table 1 summarizes the degree of correlation and the remaining errors from the multi-parameter models (r-squared and relative RMSE of the regression fits), comparing the model-predicted emission rates against those derived from the measured data. Table 1 also summarizes the accuracy and consistency of the predicted emission rates for estimating indoor formaldehyde concentrations, which are shown as the normalized RMS errors between estimated indoor formaldehyde concentrations using the predicted emission rate and the measured concentrations. Two variations on the regression models were assessed: (1) applying an eight-hour running average to the measured data, and (2) excluding hourly emission rates outside of the 5th–95th percentile range. Both of these variations improved the fitness of the multi-parameter regression, with higher r-squared values and lower relative RMSE. The improvements are expected, because both approaches intentionally eliminate the outliers of the hourly emission rate and/or decrease the noise in the data used for the regression.

While using running mean inputs and removing outliers improved emission rate predictions, they did not meaningfully improve the accuracy for estimating indoor concentrations. All three methods gave the similar results for estimating indoor concentrations using the predicted emission rates. This indicates that our original approach using one-hour average data and our assumption that the hourly emission rate was a pseudo-steady-state emission rate are reasonable.

For all three approaches, the formaldehyde concentration predictions were poor in some remaining homes. The reasons for this were unclear, but may include poor indoor mixing; unexpected ventilation (e.g., if windows or doors were opened); high emitting materials concentrated in one place (the area normalized emission assumption may not applicable); large variations in formaldehyde concentrations, temperature, *RH*, and AER between adjacent hours caused by occupant activity (the steady-state assumptions may not applicable); different loading rate and mass transport coefficients in some home (assumption of *kL* = 0.29 may not applicable); and other possible measurement errors for temperature, *RH* and AER. Illustrative examples of qualitatively poor and good predictions are shown in the time-series of measured and estimated formaldehyde concentrations in Figure 5a (poor predictions in home 46) and in Figure 5b (good predictions in home 31). In Figure 5a, some very low formaldehyde concentrations were measured without any obvious changes in other parameters, such as the air exchange rate, which might be caused by window opening by occupants. In fact, actual concentrations in Home 46 appear to be positively correlated with the ventilation rate, such that increased outside airflow leads to higher indoor concentrations. This further supports the potential for errors in the ventilation rate calculation method for this home. The good prediction in Home 31 shows agreement in both the magnitude and timing of changes in the indoor formaldehyde concentrations, with notable short-term spikes in the ventilation rate followed by temporarily reduced formaldehyde concentrations.

Our intent is to develop a working relationship that results in reasonable predictions of emission rates and the resulting indoor formaldehyde concentrations. Accordingly, we decided that the emission predictions that gave the poorest estimates on either emission rate or indoor concentration would not be used. Firstly, we removed a quarter of the homes with lowest r-square values from multi-parameter regression models using Equation (9). These homes were considered not applicable to the generalized multi-parameter regression model, because the emission rates were not well-correlated with indoor temperature, *RH*, or AER. Then, by calculating average relative difference using Equation (11) for each home, we removed the homes that had greater than 20% average relative difference, unless the average was affected by some extreme data points. This filtering process left 27 sets of estimates of predictors for the formaldehyde emission rate when using the original 1-h average approach, 27 sets for the 8-h running average approach and 26 sets for the approach excluding outliers. Most selected homes with good estimates were the same homes irrespective of the method used, but there were three to four selected homes may vary by approaches.

The estimated concentrations using the predicted emission rates based on house specific coefficients by one-hour average method for 27 selected homes with good estimators are compared to the measured formaldehyde concentrations in those same homes in Figure 6a. These results show an overall bias of prediction of less than 0.1% (slope = 0.9993). An RMSE of 3.2 µg/m^3^ (2.5 ppb) and r-square of 0.88 indicate that the general prediction is consistent, and that the fitted coefficients are also reasonable at predicting the changes in concentration as the parameters vary. For comparison, we also plot estimated concentrations using a fixed weekly average emission rate of each home versus the measured concentrations in Figure 6b. The RMSE is substantially higher (8.6 vs. 3.2) and r-square is substantially lower (0.45 vs. 0.88) when using weekly averaged emission rates compared to the dynamic emission rates predicted by temperature, *RH*, and air exchange rates. Similar comparisons are performed for the method excluding outliners (Figure 6c,d), and method using 8-h running average (Figure 6e,f). Generally, all three methods show similar improvement in predicting indoor concentrations compared to single, fixed weekly average emission rates from each home. The methods using 8-h running averages and those excluding outliers slightly underestimated overall concentrations (slopes of 0.99 and 0.98). This is expected because both alternative methods filter out very high concentrations/emission rates before applying to the regression model, and the resulting coefficient estimators are more accurate for mid-range concentrations. We’ve considered multiple guidelines that have been used for indoor air quality model evaluation [57]. The ideal IAQ model should have the observed value and predicted value plotted along the 1:1 line, with relatively smaller MSE between observations and predictions. Thus, the original 1-hr average approach was selected for further analysis.

Our overall estimate of an area normalized emission rate model for a generic house was determined by averaging the *A*, *B*, *C_st_* model coefficients for each of the selected homes. Coefficients were averaged together, because they were linearly correlated with the emission rate (i.e., the value of *A* for use in generic predictions is the average of the 27 values of *A* from the 27 homes). The resulting generic regression coefficients and variance in selected homes for each approach are shown in Table 2. The coefficients resulting from the regression model for temperature, *RH*, and equilibrium concentration are also compared to those in the previous studies. Overall, the coefficients we found from the selected homes are comparable to effects observed in previous studies.

An additional analysis was performed to evaluate the overall estimate of an area normalized emission rate model using the average *A*, *B*, *C_st_* model coefficients. The estimated concentrations in 27 selected homes using the predicted emission rates based on averaged model coefficients (*A*, *B*, *C_st_*) in Table 2 with 1-hr average method are compared to measured hourly concentrations in Figure 7a. Figure 7b shows the estimated concentrations using a single fixed floor area normalized emission rate that averaged across 27 selected homes (i.e., 23.6 μg/h/m^2^) versus the measured hourly concentrations. Significant improvement is presented when using the proposed time-varying model with averaged coefficients, which gives an overall slope of 0.98, r-square value of 0.67 and a root mean square error of 5.6 μg/m^3^ (3.9 ppb). While using an averaged fixed floor area normalized emission rate, the root mean square error between estimated and measured concentration is doubled, with a value of 11.6 μg/m^3^ (9.3 ppb) with a poor r-square value of 0.06, though the overall slope is 0.99. The overall variation of time-resolved formaldehyde concentrations in a group of homes consists of three dimensions: (1) within-home variation due to environmental condition changes during the measured week; (2) cross-home variation due to house to house environmental condition (e.g., one home may have higher indoor humidity than the other, even though same emitting materials were furnished in the two home); and (3) cross-home variation due to different emitting materials across homes. Figure 7a used the proposed model with averaged model coefficients for the whole dataset that predicted the time-resolved concentration with emission rates to account for the temperature, humidity, and air change rate variability within home and cross homes. Conversely, the results in Figure 7b, which used a fixed floor area normalized emission rate to predict time-resolved concentrations for each home in the group, did not capture any temperature, humidity, and air change rate variations. Estimated concentrations by both approaches resulted in overall slopes close to one when comparing to measured values, which indicates that the time-averaged estimated concentrations for homes by either approach would show little difference compared to averaged measured concentrations. However, using a single average the floor area normalized emission rate for all homes omits the variations of emission rates due to environmental condition changes within a home and across homes, leading to larger differences when comparing to time-resolved hourly data. We note here that the overall fit in Figure 7a is worse than Figure 6 left panel, because we used a single set of model coefficients that averaged coefficients over 27 homes to predict the time-resolved concentrations for the whole group of houses. The variation that is caused by the difference emitting materials across homes was not accounted for, leading to the estimated values having great variation across the 1:1 line. Thus, this set of averaged model coefficient would preferably be used as a predictive/comparative tool in future modeling work, rather than as a forecasting model for a specific house.

## 4. Limitations

The most important limitation of this study is the unknown bias associated with the sample of homes used to estimate the emission rates compared with any particular home (or group of homes) one may want to model. The homes where the measurements were made were relatively newly constructed (built since 2011) in California, and they were required to use low-emitting products and install mechanical ventilation. This group of 27 homes cannot be assumed to represent conditions in all homes throughout the state, let alone the US. The average temperature, humidity, and air exchange rates across the sample of homes ranged from 18–27 °C, 28–63% and 0.08–1.14 h^−1^. All regression results, therefore, must be regarded as exploratory and suggestive, and we caution their use beyond similar houses within similar environmental conditions. Our model can be improved by having a larger dataset of measurements in real homes. We are collecting more data in about 120 houses across the US. Future studies will refine the current model and improve the accuracy. The data-driven approaches will be also tested to compare the performance in real homes.

An additional source of potentially meaningful error in these emission rate estimates are the air exchange rates used to determine emission factors from measured concentrations. The air exchange rates were estimated from building leakage measurements, weather data and fan operation logging, and these estimates are subject to errors in accounting, measurement, and the models used to combine natural and mechanical flows. In addition, window and door operation cannot be ruled out as contributing to the measured concentrations, while not being reflected in air exchange rate estimates.

Finally, the accuracy of the time-resolved formaldehyde concentrations is an important source of error. Comparing one-week average concentrations for the real-time data used in this study to time averaging sensors showed across all test homes the RMSE was 6.9 µg/m^3^. The error may be larger for measuring time resolved data.

## 5. Conclusions

The intent of this work was to develop a modeling approach to determine formaldehyde emission rates in dwellings that is suitable for estimating indoor formaldehyde concentrations based on variations in indoor temperature, humidity and air change rate. This study applied an empirical model based on previous study to correlate emission rates with temperature, relative humidity, and air change rate in 63 new houses in California. Compared to the approaches using physics-based models, the method herein does not require detailed model coefficients for every emitting material to predict a dynamic whole house formaldehyde emission rate. The proposed model also provides a simplified approach to investigate emission rate variability due to environmental condition changes for a group of reprehensive homes built with low-emitting materials. In total, 27 homes with acceptable regression results were selected with resulting uncertainty in the predicted indoor formaldehyde concentrations of about 3.2 μg/m^3^ (13%). These results indicate that the simplified functional form and parameterization of the emission rate prediction is a reasonable approach. Obvious accuracy improvement was found for predicting indoor formaldehyde concentrations using the derived emission rate models, compared to using constant emission rates averaged weekly for each individual house. We caution that this likely would not be the case if this model were used to predict concentrations in any individual home, rather than using it as a predictive/comparative tool.

## Figures and Tables

**Figure 1 ijerph-19-06603-f001:**
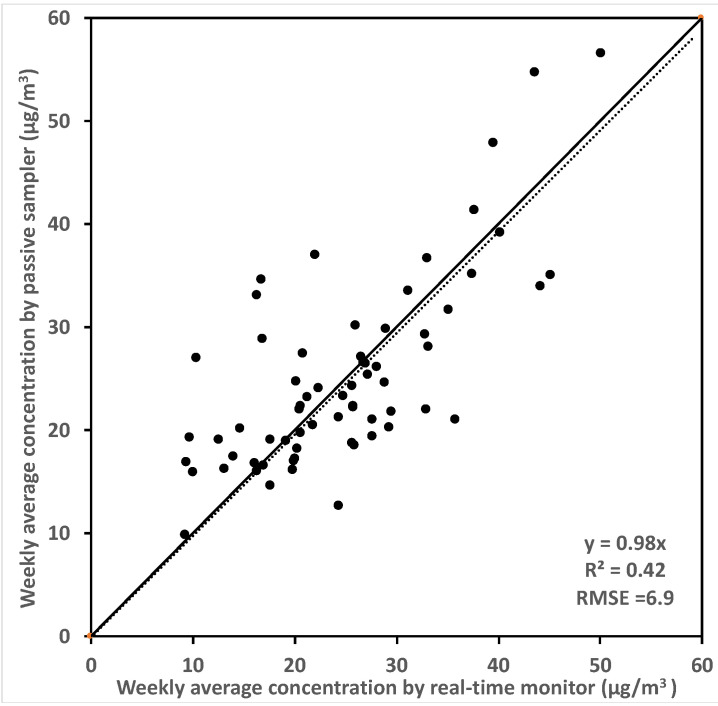
Weekly averaged formaldehyde concentrations measured by passive samplers and real-time monitors in 66 test houses, concentrations were averaged from both master bedroom and living room.

**Figure 2 ijerph-19-06603-f002:**
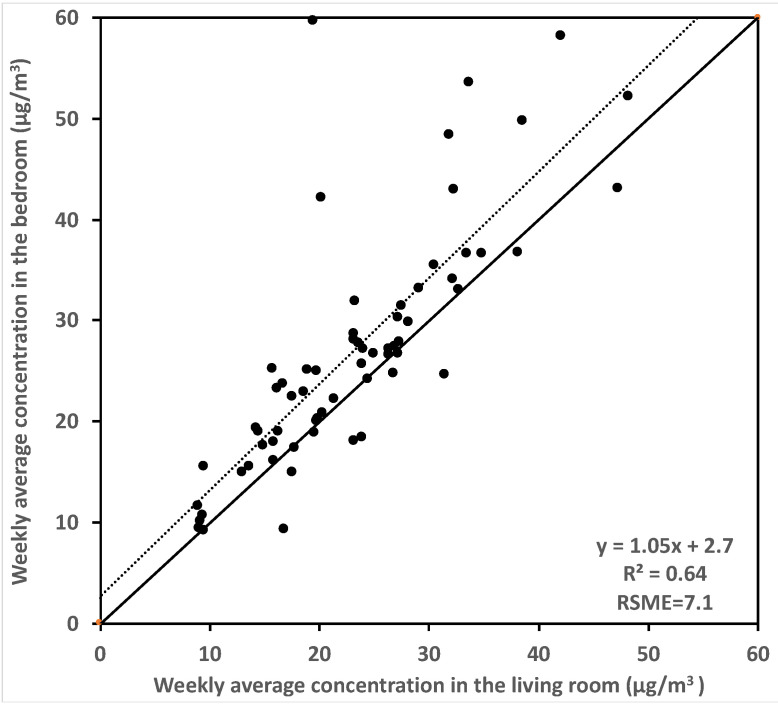
Weekly averaged formaldehyde concentrations measured in the master bedroom and living room in 63 test houses with both locations measured.

**Figure 3 ijerph-19-06603-f003:**
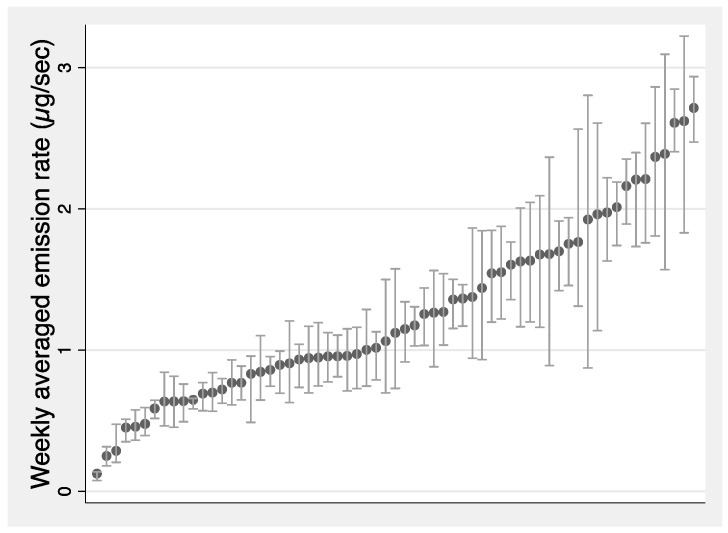
Weekly average formaldehyde emission rate (marker) for 63 new California single-family homes; shading area presents the 25th and 75th of the hourly emission rate for each home, *x*-axis rank ordered by average emission rate.

**Figure 4 ijerph-19-06603-f004:**
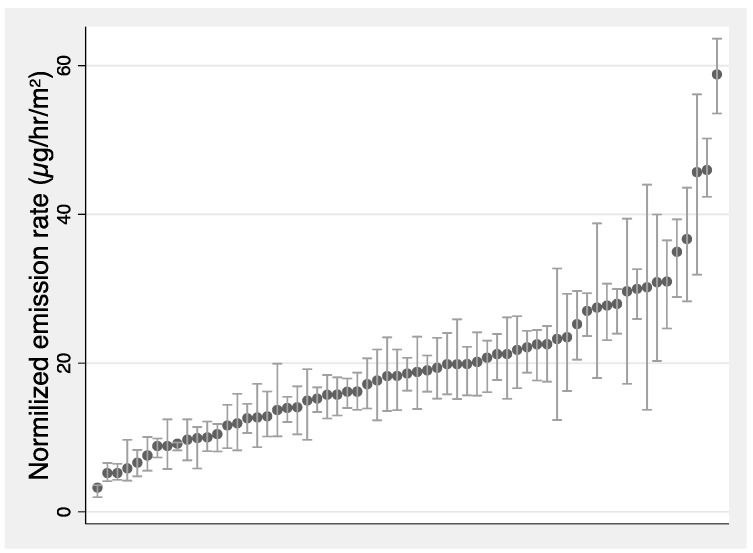
Distribution of weekly averaged formaldehyde emission rate (marker) normalized by floor area; shading area presents the 25th and 75th of the hourly normalized emission rate for each home, *x*-axis rank ordered by average normalized emission rate.

**Figure 5 ijerph-19-06603-f005:**
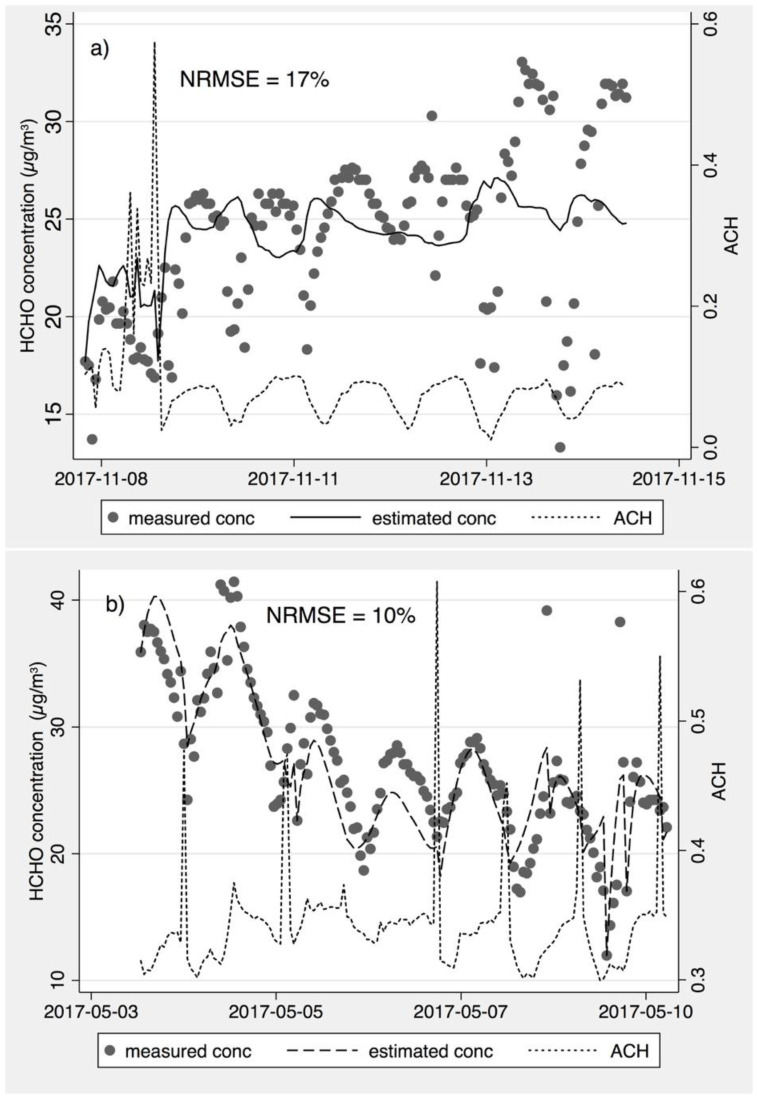
Examples of (**a**) a poor predictor and (**b**) a good predictor, by plotting time-resolved measured and estimated formaldehyde concentrations along with the air exchange rate per hour.

**Figure 6 ijerph-19-06603-f006:**
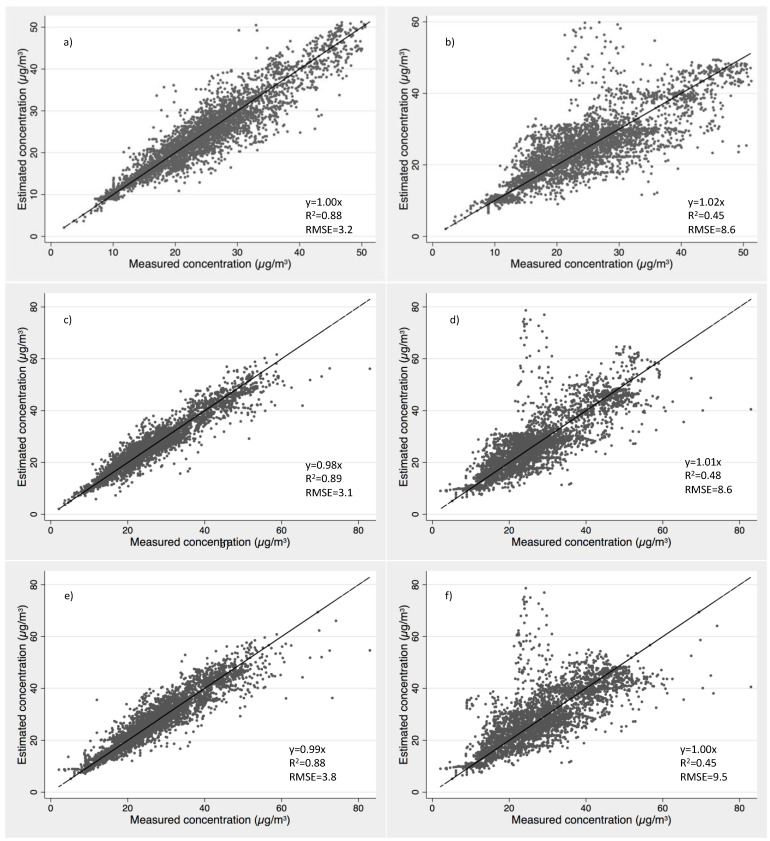
Predicted HCHO concentrations versus measured concentrations in selected houses using (**a**) emission rates predicted by the home-specific coefficients for each home using hourly averaged method, (**b**) corresponding constant emission rates by weekly average for each home using hourly averaged method, (**c**) emission rates predicted by the home-specific coefficients for each home using excluding outliner method, (**d**) corresponding constant emission rates by weekly average for each selected home using excluding outliner method, (**e**) emission rates predicted by the corresponding coefficients for each individual home using 8-hr average method, and (**f**) corresponding constant emission rates by weekly average for each selected home using 8-hr average method.

**Figure 7 ijerph-19-06603-f007:**
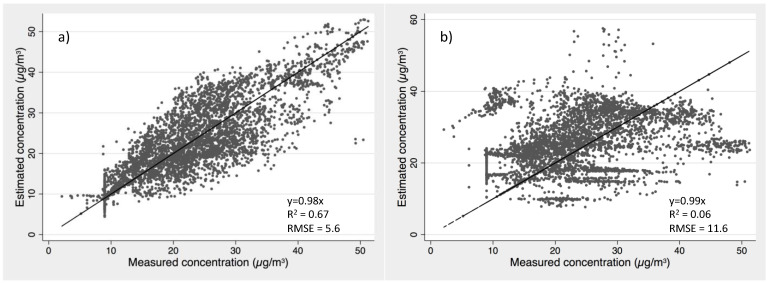
Predicted HCHO concentrations versus measured concentrations in 27 selected houses using (**a**) estimated emission rate by the proposed model with averaged coefficients across homes and (**b**) estimated emission rate by a traditional approach that using a fixed floor area normalized emission rate averaged across 27 homes.

**Table 1 ijerph-19-06603-t001:** Summary of regression performance results.

	Evaluation Criteria	1-h Average	8-h Moving Window Running Average	Use Hourly Emission Rate within 5th to 95th Percentile
Number of homes with valid coefficients		39	41	41
For predicting emission rates	R-square [Mean Median (Min–Max)]	0.88 0.91 (0.40–0.98)	0.97 0.98 (0.89–0.99)	0.94 0.96 (0.66–0.99)
	RMSE (%) [Mean Median (Min–Max)]	29% 27% (12%–81%)	15% 12% (6%–43%)	20% 19% (8%–44%)
For estimating indoor concentrations using predicted emission rates	NRMSE % [Mean Median (Min–Max)]	13% 13% (6%–24%)	11% 9% (5%–22%) ^1^ 13% 12% (6%–26%) ^2^	13% 13% (6%–25%)

^1^ NRMSE between estimated 8-h running average indoor concentration and 8-h running average measured data. ^2^ NRMSE between estimated hourly indoor concentration and hourly measured data.

**Table 2 ijerph-19-06603-t002:** Summary of regression coefficients in selected homes.

Method		1-Hour Average	8-Hour Running Average	Use Hourly Emission Rate within 5th to 95th Percentile	Reference Range from Previous Studies
Coefficient estimates [Mean Median (Min-Max)]	*A* (1/°C)	0.088 0.085 (0.015–0.253)	0.086 0.080 (0.005–0.230)	0.089 0.072 (0.007–0.203)	0.080 [27] 0.05–0.15 [48]
	*B* (1/*RH*%)	0.036 0.031 (0.009–0.076)	0.047 0.042 (0.014–0.100)	0.033 0.029 (0.006–0.078)	0.005–0.038 [48]
	*Cst* (µg/m^3^)	72.9 74.8 (24.1–117.8)	70.8 70.7 (39.6–111.1)	64.1 66.1 (37.2–90.4)	41–118 [17] 23–985 [46]

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
