# Peer review of "A Time-Varying Model for Predicting Formaldehyde Emission Rates in Homes"

_ijerph, 2022, doi:10.3390/ijerph19116603_

Round 1

Reviewer 1 Report

This is the application of a multifunctional predictive model for measuring formaldehyde levels in households, which reveals greater accuracy than the standard measurement model.

It has been carried out in few dwellings (only 65) which detracts from its external validity, however the results are striking and well documented.

The bibliography is not up to date and more recent papers should be included.

Author Response

Reviewer 1:
This is the application of a multifunctional predictive model for measuring formaldehyde levels in households, which reveals greater accuracy than the standard measurement model.
It has been carried out in few dwellings (only 65) which detracts from its external validity, however the results are striking and well documented.
The bibliography is not up to date and more recent papers should be included.

Our thanks for providing us with a conscientious review. We have updated the bibliography. We have referenced the papers in the introduction. Of note, we have included the following recent publications:

Wang, Yuanzheng, et al. "Measurement of the key parameters of VOC emissions from wooden furniture, and the impact of temperature." Atmospheric Environment 259 (2021): 118510.

Zhang, Rui, et al. "Using a machine learning approach to predict the emission characteristics of VOCs from furniture." Building and Environment 196 (2021): 107786.

Zhang, Rui, et al. "Predicting the concentrations of VOCs in a controlled chamber and an occupied classroom via a deep learning approach." Building and Environment 207 (2022): 108525.

Mohammadshirazi, Ahmad, et al. "Predicting airborne pollutant concentrations and events in a commercial building using low-cost pollutant sensors and machine learning: a case study." Building and Environment 213 (2022): 108833.

Wei, Wenjuan, et al. "Machine learning and statistical models for predicting indoor air quality." Indoor Air 29.5 (2019): 704-726.

Li, Baizhan, et al. "An investigation of formaldehyde concentration in residences and the development of a model for the prediction of its emission rates." Building and environment 147 (2019): 540-550.

Ouaret, Rachid, et al. "Spectral band decomposition combined with nonlinear models: application to indoor formaldehyde concentration forecasting." Stochastic environmental research and risk assessment 32.4 (2018): 985-997.

Bourdin, Delphine, et al. "Formaldehyde emission behavior of building materials: on-site measurements and modeling approach to predict indoor air pollution." Journal of hazardous materials 280 (2014): 164-173.

Akyüz, Ä°lker, et al. "An application of artificial neural networks for modeling formaldehyde emission based on process parameters in particleboard manufacturing process." Clean Technologies and Environmental Policy 19.5 (2017): 1449-1458.

Reviewer 2 Report

Dear authors, 

Below my comments about your paper, IJERPH-1683284. 

Best regards

============================

Overall comments

This paper present a Time-varying model for predicting formaldehyde emission rate using a multi-parameter regression model. 

1 Merits

The paper itself is well written, although somewhat descriptive. The authors have undertook a rigorous piece of data collection and have analyze information accurately. While I found the title and topic of the paper appealing, some results should be detailed.

This paper has a potential to be accepted, but some important points have to be clarified or fixed before we can proceed and a positive action can be taken.

2 References

The literature review (as well as application) focuses on physical-based models. From this point of view, it should be mentioned that it is possible to use other models, The review is then incomplete if one consider some machine learning techniques as surrogate models. The use of surrogate models for IAQ is not new, but remains insufficient to cover the state of the art. 

I suggest to add a paragraph talking about these data-driven approaches. 

Below are some of the references that are not included in the manuscript but I find relevant to this area. I would like to ask the authors if there are specific reasons why they are omitted or considered out of scope, and if there are no strong reasons to include them: 

• Akyüz et al. (2017)

• Bourdin et al. (2014) : HCHO emission behavior of building materials from on-site measurements of air phase concentration (newly built classroom). 

• Ouaret et al. (2018) : this paper focuses on developing a dynamical time series models to forecast HCHO concentrations.

• Zhang et al. (2021).

3 Main contribution, results and discussion

  1. 1. Data has already presented in previous work, many physical model have been suggested in the literature. Why using the proposed model in equation 5. How effective emitting loading rate in the house is measured in real time ?
  2. 2. R^{2} in Figure 7?
  3. 3. As suggested above, I think a more in-depth discussion of Figs. 7. I feel this is an important result since predictions were poor, and therefore it merits more discussion.
  4. 4. I think the motivations for this study need to be made clearer. The results' section should clearly explain the key limitations of prior work that are relevant to this paper. It is important to clearly explain what is new and what is not in the proposed solution. If some parts are identical, they should be appropriately cited and differences should be highlighted.
  5. 5. Overall, I think this is a study with promise, in that the authors have collected DATA in real occupational situation. However, two things would need to happen before it were ready for publication in any peer reviewed journal. 
    • (a) A much greater effort needs to be made toward describing the hypotheses and results in terms of the current state of the field.
    • (b) Additional experiments need to be conducted to help evaluate the promise of other competing hypotheses that might explain the current findings.
  6. 6. first abbreviation of OEHHA need explanation ?

Reviewer 3 Report

  1. The sentence starting at line 97 with "Those studies based on..." does not make sense, please correct;
  2. Equation (3) is not a solution to equation (2), it is a discretization of equation (2). And it is not the only one possible, there are many other choices for the discretization of a differential equation. Therefore the sentence starting at line 238 with "The time-step discretized solution..." is incorrect. I suggest to replace with something like "A discretized version of equation (2) is...".

Author Response

Reviewer 3:

  1. The sentence starting at line 97 with "Those studies based on..." does not make sense, please correct;

            We appreciate the reviewer pointing that out. We have deleted that sentence.

  1. Equation (3) is not a solution to equation (2), it is a discretization of equation (2). And it is not the only one possible, there are many other choices for the discretization of a differential equation. Therefore the sentence starting at line 238 with "The time-step discretized solution..." is incorrect. I suggest to replace with something like "A discretized version of equation (2) is...".

We have made corrections on based on the reviewer’s comments.